# Chickpea Cultivar Selection to Produce Aquafaba with Superior Emulsion Properties

**DOI:** 10.3390/foods8120685

**Published:** 2019-12-15

**Authors:** Yue He, Youn Young Shim, Rana Mustafa, Venkatesh Meda, Martin J.T. Reaney

**Affiliations:** 1Department of Chemical and Biological Engineering, University of Saskatchewan, Saskatoon, SK S7N 5A9, Canada; 2Department of Plant Sciences, University of Saskatchewan, Saskatoon, SK S7N 5A8, Canada; younyoung.shim@usask.ca (Y.Y.S.);; 3Prairie Tide Diversified Inc., Saskatoon, SK S7J 0R1, Canada; 4Guangdong Saskatchewan Oilseed Joint Laboratory, Department of Food Science and Engineering, Jinan University, Guangzhou, Guangdong 510632, China; 5Department of Integrative Biotechnology, Sungkyunkwan University, Suwon, Gyeonggi-do 16419, Korea

**Keywords:** aquafaba, chickpea, emulsifiers, egg replacer, egg-free products

## Abstract

Aquafaba (AQ), a viscous by-product solution produced during cooking chickpea or other legumes in water, is increasingly being used as an egg replacement due to its ability to form foams and emulsions. The objectives of our work were to select a chickpea cultivar that produces AQ with superior emulsion properties, and to investigate the impact of chickpea seed physicochemical properties and hydration kinetics on the properties of AQ-based emulsions. AQ from a Kabuli type chickpea cultivar (CDC Leader) had the greatest emulsion capacity (1.10 ± 0.04 m^2^/g) and stability (71.9 ± 0.8%). There were no correlations observed between AQ emulsion properties and chickpea seed proximate compositions. Meanwhile, AQ emulsion properties were negatively correlated with AQ yield and moisture content, indicating that AQ with higher dry-matter content displayed better emulsion properties. In conclusion, the emulsification properties of aquafaba are greatly influenced by the chickpea genotype, and AQ from the CDC Leader chickpea produced the most stable food oil emulsions.

## 1. Introduction

Food oil emulsions are significant components of food. Whole egg, egg yolk and egg white are typical ingredients in a range of food oil emulsions, such as mayonnaise and salad dressing, as these materials are efficient, natural emulsifiers [1] for a variety of oil/water (O/W) and water/oil (W/O) emulsions. The high emulsifying capacity of egg is related to the phospholipids (lecithin), lipoproteins (low-density lipoproteins, and high-density lipoproteins) and non-associated proteins (livetin and phosvitin) [2,3]. These proteins have amphiphilic properties and act as surface-active substances in multiphase systems, such as mayonnaise.

Unfortunately, egg products are one of the more frequent agents associated with food allergies, especially in infants and young children [4]. Egg allergens are mainly present in egg white. Ovalbumin, constituting 54% of egg white protein [4], is one of the major egg allergens [5]. In addition, egg is not suitable for consumers with special dietary restrictions, and those that cannot eat egg for religious reasons or personal lifestyle choices [6]. Moreover, egg yolk contains cholesterol (5.2% of total lipid) which is linked to cardiovascular disease.

Although a cholesterol limit is not mentioned in the 2015–2020 Dietary Guideline for Americans, it is still recommended that the elderly and people with previous incidents of heart disease limit their dietary cholesterol intake [6]. Furthermore, a segment of consumers cite environmental concerns related to egg production as a rationale to avoid egg consumption [7]. Therefore, many scientists and food processing companies are developing innovative new egg replacements to cater to a growing demand for egg alternatives.

Aquafaba (AQ) is the viscous liquid resulting from cooking chickpea seed or other legumes in water [8]. AQ has been gaining popularity since 2014, when a novel recipe blogger used the leftover liquid from a chickpea can as an egg replacement in vegan meringue [9]. Due to its desirable foaming and emulsification properties, AQ is now widely used by the vegan community as an egg replacement in many food products, such as mayonnaise, meringues and baked goods. Chickpea AQ components have been identified by Shim et al. (2018) [10]. Its application as a foaming agent has been reported in several studies [11,12]. However, the substances conferring AQ egg-similar emulsion properties have only been partially elucidated. Meanwhile, AQ qualities differ among diverse cooking conditions and legume genotypes. Therefore, chickpea cultivar selection and AQ process standardization are required to assure the quality of both AQ and AQ-based emulsions.

Based on previous studies and AQ functional properties, the central hypothesis of this research is that AQ emulsion properties not only differ among chickpea cultivars, but also have correlations with chickpea seed components and physicochemical properties. Therefore, the primary objective of this study is to prepare AQ from major chickpea cultivars and use this product to produce food oil emulsions then compare the properties of those emulsions. In addition, physicochemical properties and hydration kinetics of the different chickpea cultivars used in this study were determined to investigate possible correlations among these parameters and AQ emulsion properties.

## 2. Materials and Methods 

### 2.1. Materials

Four Kabuli chickpea cultivars (CDC Leader, CDC Orion, CDC Luna and Amit) and one Desi chickpea cultivar (CDC Consul) were generously provided by Dr. Bunyamin Tar’an from the University of Saskatchewan, Crop Development Centre (CDC, Saskatoon, SK, Canada). Seed was randomly selected and manually cleaned and freed of broken seed, dust and other foreign materials. Canola oil (purity 100%; ACH Food Companies, Inc., Terrace, IL, USA) and baking soda (NaHCO_3_; ARM & HAMMER by Church and Dwight Co., Inc., Mississauga, ON, Canada) were purchased from a local supermarket (Walmart, Saskatoon, SK, Canada). Sodium dodecyl sulfate (SDS) was purchased from GE Healthcare (Mississauga, ON, Canada). Anhydrous ether was obtained from Fisher Scientific Co. (Ottawa, ON, Canada). Sodium hydroxide (NaOH) and sodium chloride (NaCl) were purchased from Sigma-Aldrich Canada Ltd. (Oakville, ON, Canada). Concentrated sulphuric acid (H_2_SO_4_, ≥96%, *w*/*w*) and methanol were acquired from EMD Millipore Corporation (Burlington, MA, USA).

### 2.2. Fresh AQ Preparation

Chickpea seed (100 g) was washed and soaked in distilled water at a ratio of 1:4 (*w*/*w*), covered, and kept at 4 °C for 16 h [11]. Soaking water was then drained and discarded. Soaked chickpea seed (100 g) was rinsed with distilled water and then mixed with 100 mL distilled water in 250 mL sealed glass jars and cooked in a pressure cooker at 115–118 °C (an autogenic pressure range of 70–80 kPa) for 30 min. Subsequently, jars of cooked chickpeas were cooled by holding at room temperature for 24 h. Cooled AQ was drained from cooked chickpea seed using a stainless-steel strainer, then stored in a freezer (−18 °C). AQ samples from each chickpea cultivar were prepared in quadruplicate. Prior to analysis, AQ was thawed at 4 °C overnight then held at room temperature for 2 h. AQ moisture content was determined by oven drying at 105 °C overnight according to the American Association of Cereal Chemists (AACC) method 44-15.02 (AACC, 2000) [13].

### 2.3. AQ Emulsion Properties

#### 2.3.1. AQ Oil Emulsion Preparation

Freshly thawed AQ (6 g), produced from each cultivar, was mixed with 14 g canola oil using a kitchen hand mixer. The mixer was set at its maximum speed for 2 min. Canola oil was added dropwise to the AQ to produce emulsions. The emulsion type (O/W or W/O) was determined by a simple dilution test: A small amount of emulsion was dispersed into two beakers, one containing the oil phase (canola oil) and the other containing the aqueous phase (water). An easy dispersion occurs only in the continuous phase of the emulsion [14]. All emulsions prepared in this study dispersed easily in water, and were thereby confirmed to be O/W emulsions.

#### 2.3.2. Emulsion Capacity

Each AQ oil emulsion was diluted 100-fold with 0.1% SDS (*w*/*v*), and emulsion turbidity (500 nm) was calculated immediately after dilution. A UV-Vis spectrophotometer was used to determine transmittance at 500 nm. Emulsion turbidity value (*T*) was calculated using Equation (1).
(1)$$T=\frac{2.303\times A\times V}{I},$$
in which *T* is the emulsion turbidity (m^−1^), *A* is the emulsion “absorbance” measurement at 500 nm (1/transmittance), *V* is the dilution factor and *I* is the path length (0.01 m).

AQ emulsion capacity (*EC*) was determined according to Liu et al. (2016) [15]. An emulsifying activity index (*EAI*) was used as an indicator and defined by Wang et al. (2010) and Pearce and Kinsella (1978) using Equations (2) [16] and 3 [17], respectively.
(2)$$EAI=\frac{2T}{\emptyset \times C},$$
in which ∅ is the oil volume fraction of dispersed phase and *C* is the emulsifier concentration (the weight of AQ per unit volume of the aqueous phase before the emulsion is formed) [17].
(3)$$\emptyset =\frac{C-A_{1}-E\left(B-C\right)}{C-A_{1}+\frac{\left(B-C\right)\{(1+E)D_{0}-E\}}{D_{S}}}.$$

In Equation (3), *A*_1_ denotes mass of beaker; *B* is mass of beaker plus emulsion; *C* is the mass of beaker plus emulsion dry matter; *D*_0_ is the oil density; *D*_s_ is the protein solution density, and *E* is the solute concentration (mass per unit of solvent) [17]. All measurements were conducted in triplicate, and results expressed as mean ± standard deviation (SD).

#### 2.3.3. Emulsion Stability

Emulsion stability (*ES*) value was determined at room temperature. Emulsions were transferred to sealed 15 mL centrifuge tubes, which were then centrifuged at 1860× *g* for 15 min. The weight of the original emulsion before centrifugation (*F*_0_) and the emulsified layer (*F*_1_) after centrifugation were measured. The emulsion stability at room temperature was determined by Equation (4) [18].
(4)$$ES=\frac{F_{1}}{F_{0}}\times 100\%.$$

All measurements were conducted in triplicate, and results expressed as mean ± SD.

### 2.4. Chickpea Physical Properties

Hundred seed weight (*HSW*, g) was determined by randomly selecting and weighing 100 grains selected from each chickpea cultivar. Seed coat incidence (*SCI*, %) was determined by the method of Alova and Patanè (2010) with minor modification [19]. The seed coats of ten chickpea seeds were removed after soaking seed in distilled water at 4 °C for 12 h. Then the seed coat and cotyledons were dried separately at 65 °C for 4 h, and weighed each hour until a constant weight was recorded. 

Seed dimensions were determined by randomly selecting ten chickpea seeds and then using a micrometer to record the seed dimensions in three perpendicular directions. Equation (5) was used to calculate the geometric average of the diameter of an equivalent dimension (*ED*, mm).
(5)$$ED={\left(L\times W\times T\right)}^{1/3},$$
in which *L*, *W* and *T* were the major, minor and intermediate axes (mm), respectively [20].

The surface area per unit mass of seed (or, specific surface area, *SSA*, mm^2^/mg) and seed coat weight per surface area (namely seed coat thickness, *WSA*, mg/cm^2^) of a single seed were calculated based on the *ED* and *HSW* value by the following Equations (6) and (7), respectively.
(6)$$SSA=\frac{\pi \times ED^{2}\times 100}{HSW},$$
(7)$$WSA=\frac{HSW}{100\times \pi \times ED^{2}}.$$

### 2.5. Chickpea Hydration Kinetics

Hydration kinetic tests were performed by the method of Avola and Patanè (2010) [19] with minor modification. Chickpea seed was soaked at room temperature (20–22 °C) and weighed periodically to determine water uptake kinetics. Ten seeds were transferred to a 200 mL beaker, which contained 150 mL deionized water or aqueous solutions of 0.5% (*w*/*v*) NaCl or NaHCO_3_. Beakers were held at a constant temperature of 22 °C. Each hour up to the eighth hour, then at 24 h after initial imbibition, the seed was drained, and then weighed after free water was absorbed with a low-lint wiper. A clean wiper was used for each weighing to avoid contamination with solutes or water.

A two-parameter asymptotic Equation (8) was used to model water uptake kinetics (SigmaPlot 9.0; Systat Software, Inc., San Jose, CA, USA).
(8)$$H_{t}=H_{max}\times \left(1-e^{-kx}\right),$$
in which *H_t_* is hydration weight (g/seed) after soaking for time t (h), *H_max_* is the asymptote of the curve (to estimate seed weight at full hydration), k is a curve parameter that is related to the initial hydration rate (estimating *H_rate_*). All measurements were conducted in triplicate, and results expressed as mean ± SD.

### 2.6. Chickpea Chemical Properties

The moisture content of whole chickpea seed was measured by the ASAE S352.2 air oven drying method (103 °C, 72 h, 15 g) [21]. Selected whole chickpea seed was ground with a disc mill before proximate composition analysis. Analyses of crude protein, crude fat, ash and crude fibre were performed using Association of Official Analytical Chemists (AOAC) methods [22]. In brief, nitrogen content was analyzed by combustion (AOAC Method 990.03) [22] using a LECO (Saint Joseph, MI, USA) nitrogen analyzer. Protein content was calculated as nitrogen content multiplied by a conversion factor 6.25. Fat was extracted from ground samples according to AOAC method 920.39 [22] using anhydrous ether in a Soxhlet apparatus (Extraction system B-811, BÜCHI Labortechnik AG., Switzerland). Ground chickpea samples were weighed (2 g) onto filter paper which was then placed in a cellulose Soxhlet extraction thimble and washed five times with 20 mL distilled H_2_O each time. 

After drying in an oven at 102 °C for 2 h, oil was extracted over 5 h in a Soxhlet apparatus with anhydrous ether. Chickpea ash content was determined by the AOAC method 942.05 [22]. Samples were weighed (2 g) in separate, pre-weighed porcelain crucibles, and placed in a preheated furnace (600 °C) for 2 h. Crucibles were then transferred to a desiccator, cooled and reweighed. Sample weight remaining after ignition of a 2 g sample was regarded as ash content. Crude fibre content was determined by AOAC method 962.09 with minor modification [22]. Samples were digested with 1.25% (*w*/*v*) boiling H_2_SO_4_ (30 min) followed by 1.25% (*w*/*v*) boiling NaOH (30 min) and washed with methanol. Samples were then dried to a constant weight and the residue burned. Weight loss on ignition of the dried residue was regarded as crude fibre content. Carbohydrate content was determined by subtracting the total percentage of protein, fat, fibre and ash components from 100%. All measurements were conducted in triplicate, and results were expressed as mean ± SD.

### 2.7. Statistical Analysis

Three replications were used to obtain the average and SD values for all tests. Data are presented as mean ± SD (*n* = 3). Analytical results were processed with Microsoft Excel 2018. Statistics were implemented through the Statistical Package for the Social Science (SPSS) version 25.0 (IBM Corp., Armonk, NY, USA). The analysis of variance (ANOVA) and Tukey’s tests were used to evaluate the statistical significance of differences in properties and composition. Statistical significance was accepted at *p* < 0.05. The mathematical model parameters used in chickpea seed hydration kinetics measurements were estimated using a nonlinear regression procedure performed using SigmaPlot software (Systat Software Inc. San Jose, CA, USA). Model suitability was evaluated using the coefficient of determination (*R*^2^), which indicates the model predictive quality (the higher the value for *R*^2^, the better the goodness of fit, and up to a value of 1, meaning exact fit). The hydration kinetics parameters given by the nonlinear regressions were used to compare chickpea cultivars and soaking treatments, including soaking time and soaking solutions. Pearson correlation coefficients (*r*) for the relationships between all characteristics were calculated.

## 3. Results

### 3.1. AQ Produced from Different Chickpea Cultivars

AQ prepared from different chickpea cultivars showed significantly different yields and moisture contents (Figure 1). Liquid AQ yields ranged from 70.90 g/100 g seed to 107.44 g/100 g seed, with the highest yield produced by CDC Luna and the lowest by CDC Leader. AQ moisture content ranged from 92.4% to 94.2%, with the highest moisture content in AQ produced by CDC Luna and the lowest by CDC Leader. Commercially, high yield AQ with low moisture content (high dry matter content) would be of greater economic value.

Colour and turbidity of AQ varied with chickpea cultivar (Figure 2). In the current study, CDC Leader, CDC Orion, CDC Luna and Amit are Kabuli class chickpea cultivars which normally have a white to cream–yellow colour seed, while CDC Consul is a Desi class chickpea with brown to fawn colours. AQ produced from CDC Leader and CDC Orion had similar colour and high turbidity. The AQ samples were pale yellow and cloudy liquids. Whereas, AQ produced from Amit was bright yellow and cloudy. Interestingly, AQ produced by CDC Luna had the lowest turbidity, and was nearly translucent and bright yellow. Also remarkable, AQ produced by CDC Consul was a dark brown colour, and had high turbidity. This dark brown colour might arise from tannins in the CDC Consul seed coat that may have migrated to the water during cooking [23]. AQ colour is also influenced by other water-soluble molecules in chickpea seed such as pigments, vitamins and other plant secondary metabolites. Chickpea contains pigments mainly falling into the carotenoids class (β-carotene, cryptoxanthin, lutein and zeaxanthin) as well as small amounts of chlorophyll [24]. Moreover, there are also water-soluble vitamins in chickpea, such as thiamin, riboflavin, niacin, vitamin B_6_, folate and ascorbic acid [25]. Additionally, some flavonoid compounds, including anthocyanin, flavonols, isoflavones, flavonol glucosides, phlobaphenes, proanthocyanidin, leucoanthocyanidin and proanthocyanidin in the seed coat might contribute to AQ colour [26].

In general, hemicellulose [27] and cellulose [28] are probably disrupted by cooking for 30 min at 115–118 °C and autogenic pressure (70–80 kPa), leading to partial destruction of the chickpea cell wall and breaking bonds between lignin and hemicelluloses [29]. Therefore, AQ turbidity and colour is a result of the disruption of chickpea seed microstructure during cooking, leaching of organic compounds, pigments, proteins, sugars, starch and vitamins into the cooking water [30].

### 3.2. AQ Emulsification Properties

#### 3.2.1. AQ Emulsion Capacity

Today, healthier and nutritious foods are demanded by health-conscious consumers. Food oil-in-water emulsions, such as mayonnaise and salad dressing, are often avoided due to their high fat and cholesterol content. Plant-based protein fractions, including soybean and wheat proteins, are ingredients that may be used in replacing egg as emulsifiers in mayonnaise emulsion systems [31,32]. Nikzade et al. (2012) developed a combination of soy milk, gums and other stabilizers to replace egg in low cholesterol-low fat mayonnaise formulas [18]. The application and development of different ingredients in reduced fat/cholesterol salad dressing and mayonnaise have been summarized by Ma and Boye (2013) [33].

*EAI* values of AQ prepared from each of five chickpea cultivars were measured (Figure 3). *EAI* ranged from 1.10 ± 0.04 to 1.30 ± 0.05 m^2^/g with the highest *EAI* observed for AQ prepared from CDC Leader (1.30 ± 0.05 m^2^/g), while the lowest *EAI* occurred with CDC Orion (1.10 ± 0.04 m^2^/g). The *EAI*s of CDC Consul (1.21 ± 0.02 m^2^/g), CDC Luna (1.17 ± 0.07 m^2^/g), and Amit (1.20 ± 0.05 m^2^/g) were not statistically different from each other. 

#### 3.2.2. AQ Emulsion Stability

Emulsion stability of AQ from cooked whole seed of five chickpea cultivars was also investigated in this study (Figure 3). AQ emulsion stability ranged from 71.9 ± 0.8% to 77.1 ± 0.5%, with the highest emulsion stability also observed from AQ prepared by CDC Leader and Amit, while the lowest emulsion stability for AQ prepared by CDC Luna. The emulsion stability of AQ from CDC Orion (75.6%) and CDC Consul (74.7%) were not statistically different. 

As a novel water-soluble emulsifier, AQ can stabilize oil-in-water emulsions to prepare egg-free vegan food oil emulsions. Additionally, our results showed that AQ prepared from Kabuli type (CDC Leader) chickpea exerted the best emulsion capacity and stability compared to the other chickpea cultivars studied in this research. This indicates the potential for selecting CDC Leader to produce an AQ emulsifier. This difference in emulsification properties between different chickpea cultivars is probably related to the difference in the physical properties and chemical compositions of chickpea seed which affect the mass transfer to the cooking water during cooking.

To better understand the emulsion properties differences of AQ prepared from different chickpea cultivars, we studied the physiochemical properties of chickpea seed, and we tested for correlations between these properties and AQ emulsion properties. 

### 3.3. Chickpea Physical Properties

Seed coat cracking after soaking and during cooking results from splitting of the outer cell wall layers. During chickpea cooking, this seed coat works as a membrane that controls mass transfer, which would affect the composition, and therefore, the functional properties of the resulting cooking water (AQ) [8]. Seed coat physical characteristics depend on the genotype and environmental conditions (temperature, soil and moisture) at the time of seed maturity or during storage. Physical characteristics of seed from different chickpea cultivars are shown in Table 1. Significant differences were observed in *HSW*, *ED*, *SCI* and *WSA* among chickpea cultivars. *HSW* and *ED* ranged from 22.43 ± 0.08 g to 42.9 ± 0.3 g and 6.89 ± 0.5 mm to 8.49 ± 0.2 mm, respectively. CDC Leader exhibited heavier (42.90 ± 0.3 g/100 seed) and larger (8.49 ± 0.2 mm) seed compared with the other cultivars. 

CDC Orion was not significantly (*p* > 0.05) different from CDC Luna, except for *WSA* (10 ± 1 mg/cm^2^). CDC Consul exhibited the greatest *SCI* (11.2 ± 2%) and *WSA* (15 ± 1 mg/cm^2^). Amit showed the highest *SSA* (0.668 ± 0.09 mm^2^/mg). These observed values are comparable to results reported previously [19], where three Sicilian chickpea cultivars (Calia, Etna and Principe) were evaluated for their *HSW*, *ED*, *SCI*, *SSA* and *WSA*. In their study, the chickpea *HSW* value ranged from 31.3 g/100 seed to 48.8 g/100 seed. Three Sicilian chickpea cultivars had similar *ED* and *SCI* with an average of 7.8 mm and 5.78%, respectively. The *SSA* value differed among chickpea cultivars and ranged from 0.45 mm^2^/mg to 0.6 mm^2^/mg. The *WSA* value showed a wide range from 8.3 to 11.9 mg/cm^2^. These differences in the physical characteristics between different chickpea cultivars will be reflected in the seed coat behavior during soaking and cooking and might explain the variation in AQ properties. CDC Leader exhibited a good seed weight (42.90 ± 0.3 g) with the lowest *SCI* (3.89 ± 0.3%) which might explain why AQ prepared from this variety has the highest dry material (7.6%) and emulsion capacity and stability. *SCI* reflects fiber content, seed coat thickness and compactness, which is correlated with the diffusion resistance and leaching of soluble solids during soaking and cooking. The differences in the cookability of different chickpea genotypes have been reported previously which they attributed to the difference in the seed characteristics [8,19]. 

### 3.4. Hydration Kinetics

Water absorption capacity during soaking is generally related to the physical properties of the seed. Hydration of the testa and swelling of the cotyledons soften the cell walls and change tissue permeability, reduce the cooking time, and affect mass transfer to the cooking water. The relationship between soaking time and cumulative values of water uptake (Figure 4) was described by a nonlinear iterative regression method with an exponential relationship (Equation (8)). The applied model fitted the experimental data with an R^2^ that was greater than 0.94 for all cultivars. Therefore, a single curve for water uptake was used in further analysis with all data combined. Soaking processes achieve rapid water uptake (*H_rate_* = 0.38 g H_2_O g/min), and after soaking for 6 h, water absorbed reached 90% of seed dry weight. Subsequently, the water absorption rate declined until the hydrated seed weight was 2.06-fold greater than before hydration, where total hydration reached saturation at 1.06 g H_2_O g/dw (*H_max_*). Similar trends for chickpea seed hydration were described by several authors [19,34,35]. However, water content of chickpea seed exceeded 90% of total water imbibition after 4 h [19].

Statistical analysis revealed no significant differences among *H_max_*, except Amit seed which absorbed more water than the other chickpea cultivars (1.20 g H_2_O g/dw) in all soaking solutions (Table 1 and Table 2). The modeled hydration rate (*H_rate_*) of these five chickpea cultivars were mostly similar and ranged from 0.328 to 0.417 g H_2_O g/dw. Interestingly, the *H_max_* and *H_rate_* of chickpea seed soaking in NaCl solution was slightly lower compared with that obtained in deionized water. However, seed of CDC Leader, exhibited similar *H_rate_* in both deionized water and NaCl solution. Conversely, soaking seed in NaHCO_3_ solution increased *H_max_* for CDC Luna and *H_rate_* for CDC Consul. These results are partially in agreement with previous reports that the presence of salt in soaking solution results in slowed seed hydration [36], but contrast with the findings of Avola and Patanè (2010) and Clemente et al. (1998), where no effect was observed on the hydration of chickpea seed soaking in salt solutions [19,34]. 

There were two possible explanations for increased *H_max_* results in NaHCO_3_ solution: (1) the osmotic pressure gradient across membranes of cotyledon cells was decreased [37], (2) there was an interaction of carbonate ions with biopolymers in cotyledon cells which might produce molecular unfolding with a possible exposure of new sites for water binding [38].

### 3.5. Chickpea Chemical Properties

The main chemical constituents (moisture, carbohydrate, protein, fat, ash and fibre content) of different cultivars of chickpea are summarized in Table 1. The moisture content of raw chickpea seed showed significant difference among chickpea cultivars (5.29 ± 0.01% to 10.7 ± 0.1%), with the highest moisture content for CDC Consul and the lowest for Amit. Carbohydrate was the main component in all of the samples, while protein was the second most abundant component. CDC Luna had the highest carbohydrate content (67.4 ± 0.7 g 100 g/dw), followed by Amit (66.8 ± 1 g 100 g/dw) and CDC Leader (65.4 ± 2 g 100 g/dw). It is important to note that CDC Orion had the lowest carbohydrate content, which might be correlated with the lowest emulsion properties observed for AQ prepared from this cultivar. 

Protein content ranged from 18.3 ± 0.3 (CDC Luna) to 23.6 ± 0.08 g 100 g/dw (CDC Orion). The former also contained more fat (7.24 g ± 0.5 g 100 g/dw) than other chickpea cultivars, while Amit had the lowest fat content (4.10 ± 0.5 g 100 g/dw). Chickpea ash content did not differ with genotype. The mean value of ash content was 3.0 g 100 g/dw. Crude fibre content ranged from 4.32 ± 1 g 100 g/dw to 8.59 ± 0.6 g 100 g/dw. These observations are in agreement with previous studies by Xu et al. (2014), Özer et al. (2010), and de Almeida Costa et al. (2006) for chemical composition of raw chickpea seed from different chickpea cultivars [39,40,41]. In addition, Khattak et al. (2006) analyzed protein and ash content of seven Kabuli chickpea cultivars, which ranged from 18.08 to 19.22% and 2.45% to 2.94% [42], and thus, was similar to the values in this study.

### 3.6. Correlation Analysis

The overall interrelationships among chickpea physical, proximate composition (protein, fat, carbohydrate, fibre and ash), hydration characteristics, and AQ yield, emulsion capacity and stability are shown in Table 3. 

AQ yield was inversely correlated to both *ES* (*r* = −0.94*) and ash content (*r* = −0.92*). Emulsion stability was also negatively correlated to AQ moisture content (*r* = −0.91*), suggesting that AQ with higher dry matter contents have better emulsion properties. Seed coat incidence, *SSA* and *WSA* were not related to seed dimension, but *SSA* was found to have the highest negative correlation with seed weight (*r* = −0.97**). *HSW* was also positively correlated to *ED* (*r* = 0.97**), indicating that heavier and larger chickpea seeds develop a smaller surface area per unit mass. Fibre was closely correlated to *SCI* (*r* = 0.95*) and seed coat thickness (*WSA*: *r* = 0.90*). Therefore, a chickpea cultivar with low *SCI* and *WSA* will have low fibre content in the seed coat [43]. These results were observed for CDC Leader, which had the lowest *SCI* and produced AQ with superior emulsion formation and stabilization properties. 

Fat content was negatively correlated with *SSA* (*r* = −0.96*) and *WSA* (*r* = −0.90*), complementary to Gil et al. (1996) who observed a similar relationship in Desi and Kabuli chickpea [44]. In their study, fat content was also positively, and significantly, correlated to *HSW* for both chickpea classes. However, in this study, correlation between chickpea fat content and *HSW* (*r* = 0.87) was insignificant. This study supported observations made by Khattak et al. (2006), who revealed a strong, positive correlation between seed size and seed weight [42]. Moreover, they found that seed size was positively correlated with chickpea hydration capacity and protein content, as well as moisture content. However, no similar correlation was observed in this study. 

Chickpea emulsion properties have often been linked to carbohydrate content, protein content [45,46] and some phytochemicals, such as phenolics [47] and saponins [48,49]. Shim et al. [10] investigated AQ compositions recovered from commercially canned chickpea products and identified proteins present in AQ. They demonstrated that most proteins in AQ are mostly of small molecular weight (≤16.7 kDa) and many belong to the groups that include late embryogenesis abundant proteins (LEAP), dehydrins and defensin. Main carbohydrate types in AQ are simple sugars (such as sucrose and glucose), soluble and insoluble fiber including cellulose and pectin [11,12]. Importantly, the contribution of hydrophobic polysaccharides and amphiphilic phytochemicals to emulsification activity cannot be neglected. Improved rheological properties of hydrophilic polysaccharides induced steric and mechanical stabilization effects, which slowed or even prevented emulsion droplet aggregation by forming thick charged layers [50].

AQ also contains saponins, which are regarded as surfactants and emulsifiers due to their amphiphilic structure [11,49]. Chung et al. (2017) presumed that saponins could pack tightly together at the oil water interface, and thereby, effectively avoid unfavourable molecular interactions between the phases [51]. This could lower the interfacial tension to generate smaller droplets during homogenization and lead to higher emulsion stability [51]. 

Chickpea seed physicochemical properties and hydration characteristics did not correlate with AQ emulsion capacity and stability. This could be due to other factors that control the dispersal of chemical substances into AQ during cooking and the interaction between these components during cooking and storage. 

## 4. Conclusions

This study assessed the emulsion capacity and stability of AQ produced from five chickpea cultivars grown in Canada, and investigated the overall interrelationship between AQ yield and emulsion properties and chickpea physical, chemical and hydration characteristics. Our results showed that AQ produced from different chickpea cultivars illustrated significant differences in emulsion capacity and stability. AQ emulsion capacity and stability, among the five chickpea cultivars, ranged from 1.10 to 1.30 m^2^/g and 71.9 to 77.1%, respectively. Furthermore, AQ obtained from CDC Leader produced emulsions with superior emulsion capacity and stability compared to AQ prepared from other chickpea cultivars. In our study, we did not observe a significant correlation between the proximate composition of chickpea seed (carbohydrate, protein, etc.) and AQ emulsion properties (emulsion capacity and stability). However, this weak correlation did not suggest that AQ chemical components were not correlated to AQ emulsion properties, and vice versa. 

AQ composition is a complex mixture of components transferred from seed to water during cooking, in addition to other molecules produced from the interaction between these components under high pressure and temperature. Therefore, further study is needed to determine the content of other components in chickpea seed and AQ, such as saponin, fibre content and type, Maillard reaction products, and to explain the variation in AQ properties of different chickpea cultivars. In conclusion, AQ exhibits excellent emulsification properties, and is a promising emulsifier that has a great potential to be used in preparation of novel, egg-free and vegan emulsion products such as mayonnaise and salad dressing. However, the selection of chickpea genotype is required to standardize AQ production and emulsion properties.

## Figures and Tables

**Figure 1 foods-08-00685-f001:**
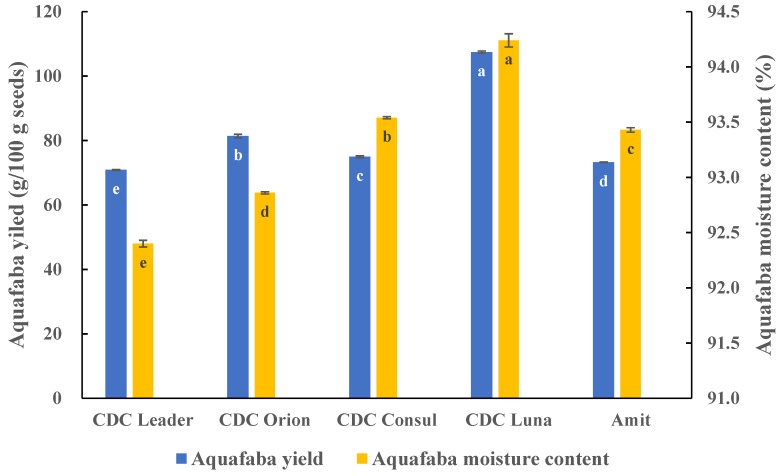
Fresh Aquafaba (AQ) yield (g/100 g seed) and moisture content (%) prepared from different chickpea cultivars from Crop Development Centre (CDC) in Saskatoon, SK, Canada. Means within the same property without a common letter (a–e) are significantly different according to Tukey’s test.

**Figure 2 foods-08-00685-f002:**
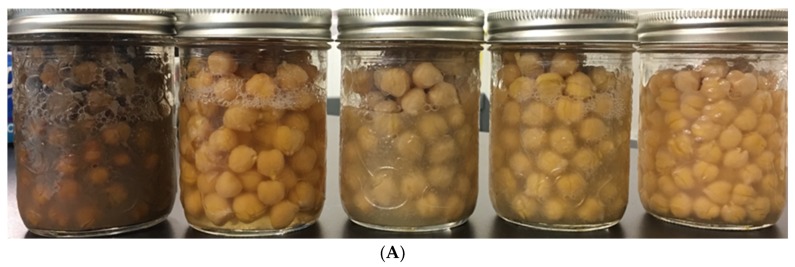
(**A**) AQ and seed of different chickpea cultivars prepared in jars. From left to right: CDC Consul, CDC Luna, CDC Orion, CDC Leader, and Amit; (**B**) AQ separated from chickpea seed. From left to right: CDC Consul, CDC Luna, CDC Leader, CDC Orion, and Amit.

**Figure 3 foods-08-00685-f003:**
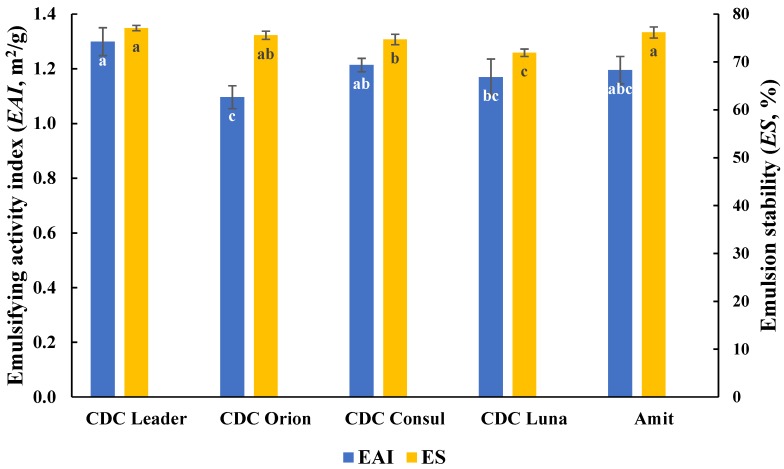
Emulsion capacity and stability of AQ prepared from different chickpea cultivars. Means within the same property without a common letter (a–c) are significantly different according to Tukey’s test.

**Figure 4 foods-08-00685-f004:**
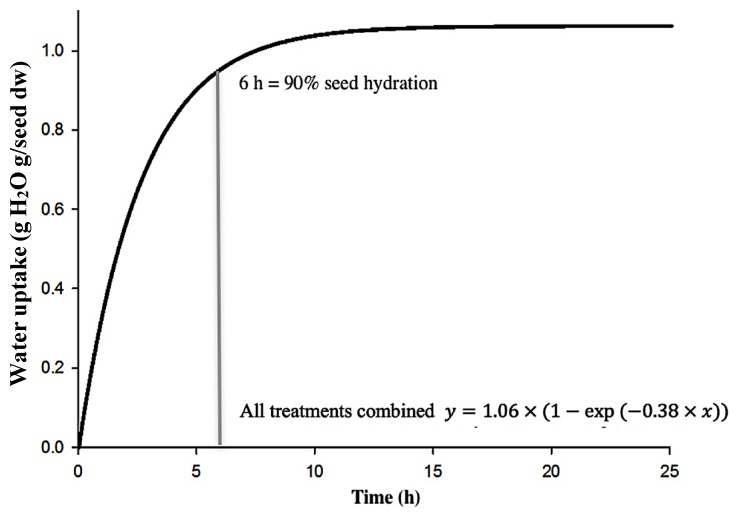
Chickpea seed water absorption kinetics. A common curve fitted all data (different chickpea cultivars and hydration solutions).

**Table 1 foods-08-00685-t001:** Physical and chemical characteristics of seed from five chickpea cultivars.

Characteristics	Unit	CDC Leader	CDC Orion	CDC Consul	CDC Luna	Amit
Physical						
Hundred seed weight, *HSW*	g	42.90 ± 0.3 ^a^	41.03 ± 0.6 ^b^	33.34 ± 0.4 ^d^	40.37 ± 0.6 ^bc^	22.43 ± 0.08 ^e^
Equivalent dimension, *ED*	mm	8.49 ± 0.2 ^a^	8.45 ± 0.3 ^a^	8.21 ± 0.2 ^a^	8.24 ± 0.1 ^a^	6.89 ± 0.5 ^b^
Seed coat incidence, *SCI*	%	3.89 ± 0.3 ^c^	6.63 ± 2 ^b^	11.2 ± 2 ^a^	5.39 ± 0.6 ^bc^	4.65 ± 0.9 ^c^
Specific surface area, *SSA*	mm^2^/mg	0.528 ± 0.03 ^b^	0.547 ± 0.04 ^b^	0.621 ± 0.03 ^ab^	0.529 ± 0.02 ^b^	0.668 ± 0.09 ^a^
Seed coat weight per surface area, *WSA*	mg/cm^2^	6.9 ± 0.4 ^d^	10 ± 1 ^c^	15 ± 1 ^a^	5.7 ± 0.2 ^d^	12 ± 2 ^b^
Technological						
Hydration capacity (t = ∞), *H_max_*	g (H_2_O g/dw)	1.036 ± 0.02 ^b^	1.073 ± 0.01 ^b^	0.9870 ± 0.03 ^b^	1.025 ± 0.1 ^b^	1.198 ± 0.03 ^a^
Hydration rate, *H_rate_*	g (H_2_O g/min)	0.3537 ± 0.01 ^ab^	0.3914 ± 0.04 ^ab^	0.3412 ± 0.09 ^b^	0.4542 ± 0.03 ^a^	0.4112 ± 0.04 ^ab^
Chemical						
Moisture	%	8.86 ± 0.07 ^c^	9.24 ± 0.08 ^b^	10.7 ± 0.1 ^a^	5.48 ± 0.02 ^d^	5.29 ± 0.01 ^e^
Carbohydrate	g (100 g/dw)	65.4 ± 2 ^ab^	61.8 ± 2 ^b^	65.2 ± 2 ^a^	67.4 ± 0.7 ^a^	66.8 ± 1 ^ab^
Protein	g (100 g/dw)	20.9 ± 0.1 ^b^	23.6 ± 0.08 ^a^	18.7 ± 1 ^c^	18.3 ± 0.3 ^c^	20.2 ± 0.1 ^b^
Fat	g (100 g/dw)	6.49 ± 0.5 ^ab^	5.96 ± 0.5 ^b^	4.64 ± 0.5 ^c^	7.24 ± 0.5 ^a^	4.10 ± 0.5 ^cd^
Ash	g (100 g/dw)	3.0 ± 0.1 ^c^	3.2 ± 0.0 ^b^	2.9 ± 0.1 ^d^	2.5 ± 0.0 ^e^	3.4 ± 0.1 ^a^
Fibre	g (100 g/dw)	4.32 ± 1 ^b^	5.36 ± 2 ^ab^	8.59 ± 0.6 ^a^	4.63 ± 1 ^b^	5.55 ± 0.8 ^ab^

Data are expressed as mean ± SD (*n* = 3). Value within rows followed by the same letter (e.g., ^a, b, c, d^) indicates no significant difference (*p* > 0.05) between varieties by Tukey’s test. CDC: Crop Development Centre (Saskatoon, SK, Canada).

**Table 2 foods-08-00685-t002:** Kinetic constants of the nonlinear regression analysis for chickpea seed hydration.

Hydration Solution	*H_max_* g H_2_O g/dw	*H_rate_* g/min	*R* ^2^
CDC Leader			
H_2_O	1.058	0.356	0.996
NaCl	1.014	0.359	0.996
NaHCO_3_	1.035	0.347	0.996
CDC Orion			
H_2_O	1.078	0.431	0.992
NaCl	1.060	0.364	0.993
NaHCO_3_	1.080	0.379	0.992
CDC Consul			
H_2_O	1.023	0.313	0.974
NaCl	0.973	0.268	0.941
NaHCO_3_	0.965	0.442	0.968
CDC Luna			
H_2_O	0.993	0.479	0.986
NaCl	0.948	0.425	0.993
NaHCO_3_	1.133	0.459	0.943
Amit			
H_2_O	1.192	0.456	0.988
NaCl	1.172	0.384	0.998
NaHCO_3_	1.230	0.393	0.988
All data combined	1.062	0.380	0.929

*H_max_*, Max hydration capacity; *H_rate_*, initial hydration rate.

**Table 3 foods-08-00685-t003:** Correlation coefficient among the sixteen physical, chemical, and hydration attributes for the five chickpea cultivars and AQ yield, emulsion capacity, and emulsion stability.

	*ES*	*EC*	*HSW*	*SCI*	*ED*	*SSA*	*WSA*	Carboh.	AQ moisture	Protein	Fibre	Fat	Ash	*H_max_*	*H_rate_*
AQ yield	−0.94 *	−0.28	0.47	0.09	0.42	−0.55	−0.50	−0.29	0.71	−0.48	−0.15	0.71	−0.92 *	−0.54	0.32
*ES*		0.40	−0.17	−0.24	−0.16	−0.24	0.25	−0.37	−0.91 *	−0.58	−0.06	−0.44	0.82	0.39	−0.37
*EC*			0.02	0.20	0.01	−0.00	−0.02	0.47	−0.34	0.39	−0.06	−0.01	−0.08	−0.17	−0.51
*HSW*				−0.10	0.97 **	−0.97 **	−0.63	−0.39	−0.28	0.24	−0.38	0.87	−0.52	−0.71	−0.19
*SCI*					0.15	0.30	0.75	0.24	0.26	−0.26	0.95 ^*^	−0.39	−0.18	−0.52	−0.62
*ED*						−0.88	−0.42	−0.48	−0.28	0.21	−0.13	0.74	−0.51	−0.83	−0.38
*SSA*							0.80	0.25	0.19	−0.21	0.57	−0.96 *	0.55	0.58	−0.03
*WSA*								−0.20	0.02	−0.04	0.90 ^*^	−0.90 *	0.45	0.12	−0.47
Carbohydrate									0.59	−0.85	−0.11	−0.01	−0.40	−0.14	0.27
AQ moisture										−0.72	0.22	0.04	−0.59	−0.07	0.43
Protein											−0.31	0.01	0.65	0.30	0.09
Fibre												−0.64	0.02	−0.30	−0.63
Fat													−0.68	−0.51	0.23
Ash														0.74	0.01
*H_max_*															0.60

AQ, aquafaba; *ES*, emulsion stability; *EC*, emulsion capacity; *HSW*, 100 seed weight; *ED*, equivalent dimension; *SCI*, seed coat incidence; *SSA*, specific surface area; *WSA*, weight of seed coat per surface area; *H_max_*, max hydration capacity; *H_rate_*, initial hydration rate. *, ** indicate significant for *p* < 0.05 and 0.01, respectively.

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
