# Peer review of "Chickpea Cultivar Selection to Produce Aquafaba with Superior Emulsion Properties"

_foods, 2019, doi:10.3390/foods8120685_

Round 1

Reviewer 1 Report

 The manuscript “Chickpea Cultivar Selection to Produce Aquafaba with Superior Emulsion Properties” describes an interesting and relevant topic.

However, the discussion lacks quality as there are no attempts to explain the observation of the study. The main objective according to the title of the manuscript is the emulsifying property. In this case, the emulsifying property results should be shown at the end of the manuscript. In this way the authors could then critically discuss their own results and link EVERY experiment with their own observations (not only with other literature).

For example line 246 ff: “Emulsification activity could be partly ascribed to proteins present which could help to decrease oil-water interface surface tension and provide electrostatic repulsion on oil droplet surfaces thereby emulsifying and stabilizing emulsions [32,33]. Importantly, the contribution of hydrophobic polysaccharides and amphiphilic phytochemicals to emulsification activity cannot be neglected.”  Table 1 contains the composition of the respective seeds, why is this Table not used to discuss the observations from Figure 3? In addition, the hydration rate can also be linked with the results.

Table 1: This analysis was performed for the chickpea seeds, but since the emulsions were prepared with AQ, these measurements should also be performed with AQ (at least protein content, polysaccharide content, fat and ash for the AQ). Then the results could provide the true correlation between the AQ composition and its functionality. what types of proteins and polysaccharides are expected in AQ? How would this contribute to the emulsifying properties observed here? Please discuss this. What is the conclusion of the study? This part is missing...

Author Response

Responses to Reviewer 1 Comments and Suggestions:

The manuscript “Chickpea Cultivar Selection to Produce Aquafaba with Superior Emulsion Properties” describes an interesting and relevant topic.

Comment: However, the discussion lacks quality as there are no attempts to explain the observation of the study. The main objective according to the title of the manuscript is the emulsifying property. In this case, the emulsifying property results should be shown at the end of the manuscript. In this way the authors could then critically discuss their own results and link EVERY experiment with their own observations (not only with other literature).

Response: We arranged the results and discussion parts in this way because the objectives of our work were to select a chickpea cultivar that produces AQ with superior emulsion properties, and to investigate the impact of chickpea seed physicochemical properties and hydration kinetics on AQ emulsion properties.

The central hypothesis of this research is that AQ emulsion properties not only differ among chickpea cultivars, but also have an interrelationship with chickpea seed composition and physicochemical properties. The first step of this study was preparing AQ from five chickpea cultivars and selecting AQ with better emulsion properties. Next step, we wanted to know if the physicochemical properties and hydration kinetics of different chickpea cultivars used in this study are correlated to AQ emulsion properties, in other words, have any influence on AQ emulsion properties.

In the results and discussion part, aquafaba basic information was introduced and compared (yield, moisture content, colour and turbidity). Then, chickpea cultivars were selected by comparing AQ emulsion properties. Next, physicochemical properties’ results of chickpea seed were analyzed. Finally, the relationship between AQ emulsion properties and the physicochemical properties of the seed was clarified and discussed by correlation analysis. Furthermore, a conclusion part has been added to emphasize the purposes and major results of this study.

Comment: For example, line 246 ff: “Importantly, the contribution of hydrophobic polysaccharides and amphiphilic phytochemicals to emulsification activity cannot be neglected.” Table 1 contains the composition of the respective seeds, why is this Table not used to discuss the observations from Figure 3? In addition, the hydration rate can also be linked with the results.

Response: We have made the requested changes and we move this paragraph (Emulsification activity could be partly ascribed to proteins present which could help to decrease oil-water interface surface tension and provide electrostatic repulsion on oil droplet surfaces thereby emulsifying and stabilizing emulsions to the Correlation Analysis (section 3.6.).

Comment: Table 1: This analysis was performed for the chickpea seeds, but since the emulsions were prepared with AQ, these measurements should also be performed with AQ (at least protein content, polysaccharide content, fat and ash for the AQ). Then the results could provide the true correlation between the AQ composition and its functionality. what types of proteins and polysaccharides are expected in AQ? How would this contribute to the emulsifying properties observed here? Please discuss this. What is the conclusion of the study? This part is missing... 

Response: We reported the physical and chemical characteristics of seed from the different chickpea cultivars to find the correlation between these properties and AQ properties. AQ emulsion capacity (EC) is based on AQ dry matter and it was determined by measuring the emulsifying activity index (EAI) which was calculated by equation (2). We addressed your comments about aquafaba composition and the correlation between the composition and the emulsifying properties and we added more information to the conclusion.

According to literature, AQ emulsion properties were linked to some components such as carbohydrate, protein, phenolics and saponins. In manuscript line 345: “Chickpea emulsion properties have often been linked to carbohydrate content, protein content [44,45], and some phytochemicals such as phenolics [46] and saponins [47,48].”

In this study, our hypothesis is that “AQ emulsion properties have interrelationship with chickpea seed components and physicochemical properties”. Therefore, in this study we only determined chickpea seed proximate compositions (protein, fat, carbohydrate, fibre, and ash) rather than AQ compositions, and compared it with AQ emulsion properties. Seeds are cooked under the same conditions when preparing AQ. If AQ emulsion properties are correlated to chickpea component content (protein, carbohydrate content etc.), then chickpea cultivar selection would be much easier. For example, if AQ emulsion properties are positively correlated to chickpea protein content, we would only need to choose the chickpea cultivar with higher protein content to make emulsion for further study.

However, in lines 370-375 “AQ obtained from CDC Leader produced emulsions with superior emulsion capacity and stability compared to AQ prepared from other chickpea cultivars. In our study, we did not observe a significant correlation between the proximate composition of chickpea seed (carbohydrate, protein, etc.) and AQ emulsion properties (emulsion capacity and stability). However, this weak correlation did not suggest that AQ chemical components were not correlated to AQ emulsion properties, and vice versa.”

Also, proteins and polysaccharides types are added in lines 345-354. Their contributions to the emulsifying properties are introduced.

Reviewer 2 Report

The manuscript entitled “Chickpea Cultivar Selection to Produce Aquafaba with Superior Emulsion Properties” deals with the study of emulsions stabilized by the surface active agents contained in the AQ from different chickpea cultivars. The Authors found that AQ from CDC Leader cultivar had the greatest emulsive capacity and stability and that there was no correlation between AQ and seed proximate composition. The emulsiove properties of AQ were also negatively correlated with moisture content.

The manuscript is interesting but there are some points to be clarified:

Page 2 line 86-86,what does the sentence “Each chickpea cultivar was used to produce AQ in this way four times” means?

Page 2 line 91 “2.3. AQ from Five Chickpea Cultivars”. I think the title of this paragraph should be modified since paragraphs 2.3.1 and 2.3.2 refer to the emulsion properties of AQ.

Page 3, from line 93 the composition of emulsions is unclear. AQ 6g is referred to dry matter or to aqueous dispersion of dry extract of AQ? What is the water content of the emulsions. If in each emulsion there are 6g of AQ (water included) and 14 g of canola oil I expect a water in oil emulsion, not oil in water as reported.

In the captions of the figures 1 and 3, please include what the  letters on the columns stand for.

The results and the discussion relative to the emulsion stability are incoherent. The authors compare their emulsion stability results to those of reference [16] where different egg-replacers (not AQ) are used in the mayonnaise production. The Authors conclude that …(page 8 line 262)”…AQ has better emulsification effects than some existing commercial egg replacers when used as emulsifiers in mayonnaise” but they do not prepare mayonnaise.

The composition of salad dressing like mayonnaise is quite different from the one reported in this manuscript, so I think it is not appropriate and rigorous to compare the emulsifiers performances in different preparations.

Author Response

Manuscript ID foods-657464 entitled “Chickpea cultivar selection to produce aquafaba with superior emulsion properties”

Thank you for your patience. We have revised our manuscript (Manuscript ID: foods-657464) according to the reviewers’ comments/suggestions and we would like to thank all reviewers for their critical feedback in making this manuscript more polished. We have listed the reviewers’ comments and answered them in sequence. In addition to the changes made to the manuscript as recommended by the reviewers, we have also pursued to improve the manuscript’s structure. We appreciate the reviewers’ thoughtful comments and critiques and hope these addresses help in improving the overall quality of this manuscript for publication.

Responses to Reviewer 2 Comments and Suggestions:
The manuscript entitled “Chickpea Cultivar Selection to Produce Aquafaba with Superior Emulsion Properties” deals with the study of emulsions stabilized by the surface-active agents contained in the AQ from different chickpea cultivars. The Authors found that AQ from CDC Leader cultivar had the greatest emulsion capacity and stability and that there was no correlation between AQ and seed proximate composition. The emulsion properties of AQ were also negatively correlated with moisture content. The manuscript is interesting but there are some points to be clarified:

Comment: Page 2 line 86-87, what does the sentence “Each chickpea cultivar was used to produce AQ in this way four times” means?

Response: This sentence has been revised. It means that we prepared four AQ samples for each of the chickpea cultivar using the same cooking way (in quadruplicate) on line 96 on page 5.

Comment: Page 2 line 91 “2.3. AQ from Five Chickpea Cultivars”. I think the title of this paragraph should be modified since paragraphs 2.3.1 and 2.3.2 refer to the emulsion properties of AQ.

Response: This title has been revised. A new title is “AQ Emulsion Properties” on line 101.

Comment: Page 3, from line 93 the composition of emulsions is unclear. AQ 6g is referred to dry matter or to the aqueous dispersion of the dry extract of AQ? What is the water content of the emulsions? If in each emulsion there are 6g of AQ (water included) and 14 g of canola oil I expect water in oil emulsion, not oil in water as reported.

Response: AQ (6 g) is referred to aqueous dispersion of dry extract of AQ or in another word AQ liquid. The water content of AQ is (92-94%, shown in Figure 1), therefore, the water content of the emulsions is 6*93%/(6+14)*100% = 27.9%.

In order to get oil in water emulsion, canola oil has to be “added dropwise to the AQ” (line 103 in 2.3.1) during blending with a mixer rather than mixing all AQ and oil and blending together (this will result in water in oil emulsion).

Aquafaba plays the role of a surfactant and it is soluble in water, which would lead to the formation of oil-in-water (o/w) emulsion (Winsor I emulsion) because of R < 1 in the Winsor`s ratio.

Mayonnaise is also o/w emulsions even though large quantities of oil are emulsified in a relatively small amount of water.

Comment: In the captions of the Figures 1 and 3, please include what the letters on the columns stand for.

Response: Figures 1 and 3 have been revised (lines 518 and 524).

Comment: The results and the discussion relative to the emulsion stability are incoherent. The authors compare their emulsion stability results to those of reference [16] where different egg-replacers (not AQ) are used in the mayonnaise production. The Authors conclude that ...(page 8 line 262)”...AQ has better emulsification effects than some existing commercial egg replacers when used as emulsifiers in mayonnaise” but they do not prepare mayonnaise.

Response: This part has been revised. Reference [16] was deleted and more explanations for AQ emulsion stability were added lines 244-253 and lines 355-359.

Comment: The composition of salad dressing like mayonnaise is quite different from the one reported in this manuscript, so I think it is not appropriate and rigorous to compare the emulsifiers' performances in different preparations.

Response: The comparison of emulsifiers in mayonnaise and AQ has been removed.

Round 2

Reviewer 1 Report

All issues have been adressed

Author Response

Thank you.

Reviewer 2 Report

In the revised version of the manuscript the authors have addressed some of my concerns, however they are not convincing on the structure of the emulsion they proposed especially for the theory they recalled.

In the cover letter the authors justify the oil in water structure on the basis of a Winsor I emulsion. The Winsor classification  is applied to the study of microemulsions that are systems very different from the emulsions for their stability (microemulsions are more stable) and for the diameter of the dispersed phase (few nanometers, 5-20nm, for the microemulsions) . Microemulsions are isotropic, transparent systems and the Winsor I is formed by a o/w microemulsion having a layer of oil stratified on top. This is not the system the authors described.

In order to clarify the structure of their systems, I suggest the authors to add a figure showing to aspect of  the emulsions through an optical microscope. If needed they can dissolve a dye in the oil or in  the water phase so the visualization of the structure can be clearer. At the same time the diameter of the dispersed phase could be determined.

Author Response

Thank you for your patience. We have revised our manuscript (Manuscript ID: foods-657464) according to reviewer 2’s comments/suggestions and would like to thank all reviewers for their critical feedback in making this manuscript more polished. We have listed reviewer 2’s comments and answered them in sequence. In addition to the changes made to the manuscript as recommended by the reviewers, we have also tried to improve the manuscript’s structure. We appreciate the reviewers’ thoughtful comments and critiques and hope this response addresses the overall quality of this manuscript for publication.

Responses to Reviewer 2 Comments and Suggestions:
In the revised version of the manuscript the authors have addressed some of my concerns, however, they are not convincing on the structure of the emulsion they proposed especially for the theory they recalled.

Comment: In the cover letter the authors justify the oil in water structure on the basis of a Winsor I emulsion. The Winsor classification is applied to the study of microemulsions that are systems very different from the emulsions for their stability (microemulsions are more stable) and for the diameter of the dispersed phase (few nanometers, 5–20 nm, for the microemulsions). Microemulsions are isotropic, transparent systems and the Winsor I is formed by an O/W microemulsion having a layer of oil stratified on top. This is not the system the authors described.

In order to clarify the structure of their systems, I suggest the authors to add a figure showing to aspect of the emulsions through an optical microscope. If needed they can dissolve a dye in the oil or in the water phase so the visualization of the structure can be clearer. At the same time, the diameter of the dispersed phase could be determined.

Response: The authors submit that they never suggested that the opaque emulsions prepared from AQ are Winsor I emulsions. We did a test to determine if the emulsions had a continuous oil or water phase. Others have prepared O/W emulsion by adding canola oil “dropwise to the AQ” during blending. If mixing all AQ and oil at once, the result is a W/O emulsion. In order to verify the system O/W structure, we conducted a simple dilution test. A small amount of emulsion was dispersed into two beakers, one containing the oil phase (canola oil) and the other containing the aqueous phase (water), and easy dispersion being performed only in the continuous phase of the emulsion. We found that AQ emulsions dispersed easily in water and determined that the system is an O/W emulsion. This procedure and result have been added to the manuscript (Lines 104–108).
